# Detection and Genetic Characterization of Astroviruses in Brain Tissues of Wild Raccoon Dogs

**DOI:** 10.3390/v15071488

**Published:** 2023-06-30

**Authors:** Su-Beom Chae, Chang-Gi Jeong, Jun-Soo Park, Eun-Jee Na, Jae-Ku Oem

**Affiliations:** Laboratory of Veterinary Infectious Disease, College of Veterinary of Medicine, Jeonbuk National University, Iksan 54596, Republic of Korea; cotnqja23@gmail.com (S.-B.C.); jcg0102@gmail.com (C.-G.J.); spinyang@naver.com (J.-S.P.); ejna1212@naver.com (E.-J.N.)

**Keywords:** astrovirus, wild raccoon dog, neurotropic-astrovirus

## Abstract

Astroviruses (AstVs) have been detected in a wide range of animal species, including mammals and birds. Recently, a novel AstV associated with neurological symptoms has been detected in the brains of some mammals. Raccoon dog AstV has been reported recently in China. However, there have been no reports in South Korea. Therefore, the present study aimed to detect and genetically characterize AstVs in the intestine and brain tissues of 133 wild raccoon dogs collected in Korea between 2017 and 2019. Of the seven raccoon dogs, AstVs were detected in six intestinal tissues and four brain tissues. Analysis of the capsid protein amino acid sequences of raccoon dog AstVs detected in Korea revealed a high similarity to canine AstVs, suggesting possible interspecies transmission between raccoon dogs and dogs. Phylogenetic and capsid protein amino acid sequence analysis of raccoon dog AstVs detected in the brain the 17-148B strain belonging to the HMO clade and exhibiting conserved sequences found in neurotropic AstVs (NT-AstVs), indicating their potential as NT-AstVs. However, the pathogenicity and transmission routes of the raccoon dog AstV detected in Korea have not yet been elucidated, so further research and continued surveillance for AstV in wild raccoon dogs are needed.

## 1. Introduction

AstVs are a small non-enveloped type of virus belonging to the family *Astroviridae* [1]. As a positive-sense, single-stranded RNA virus, its viral genome is 6.8–7.9 kb in length and consists of three overlapped open reading frames (ORFs) known as ORF1a, ORF1b, and ORF2a, encoding non-structural proteins, RNA-dependent RNA polymerase (RdRp), and viral capsid protein, respectively [2].

Since their discovery in a child in 1975, AstVs have been detected in various hosts, sich as pigs, turkeys, cattle, chickens, mink, cats, and dogs, and novel AstVs continue to be detected in various animals [3], including reptiles, fish, and amphibians [4]. Since the 1970s, there has been a steady increase in the number of publications on AstV, and the discovery of novel AstV strains through next-generation sequencing (NGS) technologies and the potential for zoonotic transmission have further increased the interest in AstV [2,5,6].

AstVs are taxonomically classified into two genera based on the complete amino acid sequence of the capsid protein: *Mamastrovirus* (*MAstV*) *1–19*, which generally infects mammalian species, and *Avastrovirus* (*AAstV*) *1–3*, which generally infects avian species [3]. AstV is primarily known to infect the gastrointestinal tract of animals, causing gastroenteritis and being associated with symptoms such as diarrhea [4]. AstV causes mainly asymptomatic infections or mild gastroenteritis in mammals, and causes various pathologies in birds, including enteritis, hepatitis, and nephritis [4]. Animals infected with AstV can excrete the virus in their feces, contaminate water and food, and infect other animals via the fecal–oral route, which is the primary route of transmission [7].

Known to infect the gastrointestinal tract, AstV has also been identified to infect the livers and kidneys of avian species, causing symptoms, confirming that AstV can infect organs other than the gastrointestinal tract [8,9]. In 2010, novel AstVs associated with neurological symptoms were detected in the brains of humans and mink, and these neurotropic AstVs (NT-AstVs) are progressively being identified in the brains of other mammals, including sheep, cattle, pigs, and alpacas [10,11]. Animals infected with NT-AstV in the brain show neurological signs such as tremors and limb paralysis, and generally have non-suppurative encephalitis [12]. The mechanism for the onset of neurological signs in NT-AstV infection remains unclear; however, some factors, such as underlying disease, co-infections, or stress, have been reported to probably affect some animals [11]. The transmission of NT-AstV is also still unclear, but cases wherein NT-AstV has been detected in the gastrointestinal tract and feces suggest that NT-AstV may be transmitted by the fecal–oral route. [13,14]. Based on their phylogenetic analysis, NT-AstVs are divided into two clades, the HMO (human–mink–ovine/bovine) clade and the MLB (Melbourne) clade [12]. The NT-AstVs detected in humans have been identified in both the HMO and MLB clades, whereas NT-AstVs detected in animals to date all belong to the HMO clade [11]. NT-AstV strains belonging to the HMO clade caused similar clinical symptoms and histopathological lesions, and had higher neurotrophic potential than AstVs belonging to other groups [12].

Based on many previous studies, raccoon dogs (*Nyctereutes procyonoides*) are known to be vectors for the transmission of infectious diseases, such as canine distemper virus (CDV), rabies, helminths, and tick-borne diseases between livestock and wildlife in Europe and Asia [15,16]. In raccoon dogs, AstV was first detected through metagenomic analysis in China in 2021 and 2022 [17,18]. Some raccoon dog AstV strains are clustered with *MAstV 5,* to which canine AstV belongs, but the correlation between the canine and raccoon dog AstVs is unknown [17]. Raccoon dogs are increasing in population due to their adaptability to various environments and their lack of natural enemies, and their potential to invade human habitats has led to increased interest in pathogens for which they are hosts [19,20]. Raccoon dog-related diseases reported in Korea are mainly focused on several pathogens such as CDV and rabies, and AstV has never been reported in raccoon dogs [15]. Therefore, the aim of this study was to investigate the presence and genetic characterization of AstV in 133 wild raccoon dogs collected in Korea between 2017 and 2019.

## 2. Materials and Methods

### 2.1. Sample Collection

AstV detection was conducted using brain and intestinal tissue samples obtained from wild raccoon dogs inhabiting South Korea. These raccoon dogs were provided by the National Institute of Wildlife Disease Control and Prevention. A total of 133 brain tissues and 77 intestinal tissues collected from 133 wild raccoon dogs between 2017 and 2019 were used for AstV testing. Carcasses of raccoon dogs found dead due to traffic accidents or diseases were stored and transported in a freezer before necropsy. Each tissue sample was stored at −70 °C until nucleic acid extraction.

### 2.2. Viral RNA Extraction and cDNA Synthesis

Each tissue sample was homogenized in 1 mL of 1× phosphate-buffered saline (PBS) at a 10% volume. After homogenization, the samples were centrifuged at 10,000× *g* for 10 min at 4 °C. The resulting supernatant was filtered using a 0.45 μm filter (BioFACT, Daejeon, Republic of Korea). The filtered supernatant was treated with 10× DNase buffer (Roche, Mannheim, Germany) and DNase I (Roche, Mannheim, Germany) at a final volume of 300 μL and incubated at 37 °C for 2 h. RNA extraction was performed using a hybrid protocol combining TRIzol reagent (Thermo Fisher Scientific, Waltham, MA, USA) and RNeasy mini kit (Qiagen, Hilden, Germany) [21]. The extracted RNA was eluted with 30 μL of elution buffer. Viral RNA was denatured at 65 °C for 3 min, followed by cDNA synthesis using random hexamers and oligo dT primers with an AccuPower^®®^ RocketScript™ RT Master Mix with RNase H Minus (Bioneer, Daejeon, Republic of Korea).

### 2.3. Astrovirus Screening and ORF2 PCR

To detect AstVs at the molecular level, pan-AstV screening was performed targeting a 422 bp partial RdRp gene, which is known to be the most conserved within AstVs following the method described by Chu [22]. PCR amplicons were purified using a PCR clean-up kit (Cosmogenetech, Seoul, Republic of Korea) and sequenced using commercial sequencing services (Macrogen, Seoul, Republic of Korea). Nested PCR was carried out to amplify the complete ORF2 capsid sequence of 2700 bp. This involved using a stem-loop-2-like motif (s2m) reverse primer and a gene-specific primer (GSP) within the RdRp partial sequence (Table 1). The nested PCR protocol consisted of an initial denaturation step at 94 °C for 5 min, 35 cycles of denaturation at 94 °C for 1 min, annealing at 57 °C for 1 min, and extension at 72 °C for 3 min, followed by a final extension step at 72 °C for 10 min. ORF2 PCR products were purified using a PCR clean-up kit (Cosmogenetech, Seoul, Republic of Korea) and sequenced using the BT Seq TM-Standard service (Celemics, Seoul, Republic of Korea).

### 2.4. Genetic Analysis of Astrovirus Sequences

Phylogenetic analysis was conducted based on partial RdRp nucleotide and complete ORF2 amino acid sequences. Multiple sequence alignments of RdRp nucleotide and ORF2 amino acid sequences were generated using Clustal Omega. Phylogenetic trees were constructed using the neighbor-joining (NJ) algorithm with 1000 bootstrap replicates in MEGA X software version 10.2.6. The variability analysis of complete capsid protein amino acid sequence of AstVs was performed using an online Protein Variability Server (http://imed.med.ucm.es/PVS/ accessed on 25 April 2023).

## 3. Results

### 3.1. Detection of Raccoon Dog Astrovirus

Brain and intestinal tissues from 133 wild raccoon dogs collected in Korea from 2017 to 2019 were tested, and seven raccoon dogs were confirmed positive for AstV, with AstV RNA detected in six intestinal tissues and four brain tissues (Table 2). The overall prevalence of AstVs in wild raccoon dogs in South Korea was 5.2% between 2017 and 2019, and 7.8% and 3.0% of these occurrences were shown in the intestine and brain samples of raccoon dogs, respectively. Of the six raccoon dogs with AstV detected in intestinal tissue, raccoon dogs 17-148, 17-153, and 18-038 also had AstV detected in brain tissue. No AstV was detected in the intestinal tissue of raccoon dog 17-157, which had AstV detected in its brain tissue. Of the raccoon dogs with AstV detected, no other pathogens were identified, except for detecting *Sarcoptes scabiei* in raccoon dog 18-038.

### 3.2. Genetic Analysis of Partial RdRp Sequences

Partial RdRp nucleotide sequences were obtained from each tissue identified as positive for AstVs by pan-AstV screening (accession numbers: OR043647-OR043656). The obtained partial RdRp sequences were BLAST searched to identify similar sequences in the NCBI database (Table 3). As a result, AstV detected in the brain and intestine of raccoon dog 17-148 (17-148B and 17-148I) showed the highest similarity (90.1%) to the AstV detected in Ailurus fulgens from China. AstV detected in the brain or intestine of raccoon dogs 17-153, 17-157, 17-162, 17-165 and 18-038 (17-153B, 17-153I, 17-157B, 17-162I, 17-165I, 18-038B and 18-038I) had the highest similarity (93.2% to 97.5%) to the canine AstV strain HUN/126 reported in Hungary. AstV detected in the intestine of raccoon dog 18-026 (18-026I) showed the highest similarity to chicken AstV at 76.7%. A comparison of the raccoon dog AstVs detected in Korea with those previously reported in China showed differences, with similarities ranging from 43.3% to 89.3%. The phylogenetic analysis of partial RdRp nucleotide sequences showed that these AstVs detected in wild raccoon dogs in Korea formed a distinct cluster with canine AstVs, except for strains 17-148B, 17-148I, and 18-026I (Figure 1).

### 3.3. Genetic Analysis of Complete Capsid Protein Amino Acid Sequences

To clarify the genetic characteristics of AstV detected in wild raccoon dogs in Korea, we performed a similarity analysis on the amino acid sequence of the capsid protein obtained through the ORF2 gene (Figure 2). Amino acid homology among Korean raccoon dog AstV strains ranged from 16.6% to 100%, showing significant differences in some strains. The amino acid capsid protein sequence identity between AstVs detected in the brain and in the intestine of the same raccoon dog was 100% between 17-148B and 17-148I, 91.1% between 17-153B and 17-153I, and 99.7% between 18-038B and 18-038I. The capsid protein amino acid sequence similarity between 17-157B and 17-153I strains, which showed 100% nucleotide sequence identity to the partial RdRp gene, was found to be 90.9%. In addition, raccoon dogs AstVs 17-148B, 17-148I, and 18-026I showed significant differences in capsid protein amino acid sequences, with 16.6% to 26.3% similarity compared to other raccoon dog AstVs in Korea.

Based on BLAST searches of capsid protein amino acid sequences, raccoon dog AstVs 17-153B, 17-153I, 17-157B, 17-162I, 17-165I, 18-038B, and 18-038I showed the highest similarity to canine AstV HUN/126 identified in Hungary, as did the partial RdRp sequence (74.9% to 83.5%) (Table 4). Raccoon dog AstV 17-148B and 17-148I showed the highest similarity (91.4%) to canine AstV HUN/8 identified in Hungary in terms of capsid protein amino acid sequence. Raccoon dog AstV 18-026I showed 96.3% capsid protein amino acid sequence similarity to chicken AstV, which is different from the results seen with the partial RdRp gene. The capsid protein amino acid sequence similarity between raccoon dog AstV strains reported in China and raccoon dog AstVs detected in Korea showed significant differences, ranging from 13.1% to 59.1%.

An analysis of capsid protein variability and similarity between Hungary canine AstV strains, raccoon dog AstVs reported in China, and raccoon dog AstVs detected in Korea revealed that strains 17-148B and 17-148I showed high similarity to canine AstV strain HUN/8 in both core and spike protein regions (97.5% and 93.5%, respectively) (Figure 3A). Raccoon dog AstV strains clustered with MAstV5 showed significant similarity to canine AstV strain HUN/126 in the core protein region (98.0–99.1%), although they differed in the spike protein region (46.4–66.0%) (Figure 3B).

Phylogenetic analyses of the complete capsid protein amino acid sequences of the AstVs revealed that raccoon dog AstVs 17-153B, 17-153I, 17-157B, 17-162I, 17-165I, 18-038B and 18-038I belong to *MAstV 5,* along with canine AstVs (Figure 4). Raccoon dog AstVs 17-148B and 17-148I differed from other raccoon dog AstVs detected in Korea and were placed in the unclassified *MAstV* group, along with the canine AstV strain HUN/8, mink AstV, and raccoon dog AstV 3 (Figure 4). Raccoon dog AstV 18-026I, which showed similarities to chicken AstVs, was not grouped with *MAstV* and was placed in the group of *AAstV*, which are known to infect birds, along with chicken AstVs (Figure 4).

For the brain tissue of raccoon dog samples in which AstV was detected, histopathological examination results could not be obtained due to tissue damage caused by carcass cryopreservation. Therefore, genetic analysis was performed on the AstV strains (17-148B, 17-153B, 17-157B, 18-038B) detected in the brain tissue to investigate the potential of NT-AstV. Phylogenetic analysis revealed that strain 17-148B belongs to the HMO clade, a clade of NT-AstV (Figure 4). Furthermore, capsid protein amino acid sequence analysis confirmed the presence of a conserved Q(I/L)QxR(F/Y) motif sequence in the HMO clade of NT-AstV (Figure 5). However, the other AstV strains (17-153B, 17-157B, 18-038B) detected in the brain tissue of raccoon dogs did not belong to either clade of NT-AstV, and conserved protein sequences such as the Q(I/L)QxR(F/Y) motif sequence were not detected.

## 4. Discussion

Previous studies have shown that the prevalence of AstV in wildlife, including rodents, bats, and birds, is approximately 5–11% [24,25]. The reported prevalence in raccoon dogs in China was also 6.3%, which is similar to the prevalence in other wildlife [18]. The prevalence of AstVs in intestinal samples from wild raccoon dogs in Korea was found to be 7.8%, similar to the prevalence in Chinese raccoon dogs. Additionally, the prevalence of AstV in brain samples from wild raccoon dogs was found to be 3%. However, most studies of NT-AstV have focused on humans or domestic animals such as pigs, cattle, sheep, and mink [10,11]. As a result, there is a lack of information on the detection or study of NT-AstV in wildlife, so data on the prevalence could not be compared. Therefore, there is a need to conduct continuous monitoring of AstVs in wildlife, such as raccoon dogs, to enable the detection and assess the prevalence of NT-AstV in wildlife populations.

Previous studies have reported the possibility of interspecies transmission of AstV infecting different host species [26]. In this study, similarity and phylogenetic analyses of the partial RdRp gene of raccoon dog AstV detected in Korea showed differences in nucleotide and amino acid similarity with raccoon dog AstV strains reported in China. However, raccoon dog AstV detected in Korea was found to be very similar to AstVs detected in other species, such as dogs and chickens. These results suggest the possibility of the interspecies transmission of AstV between wild raccoon dogs and other host species in Korea. However, since the partial RdRp nucleotide sequence was relatively short, about 400 bp in length, we performed sequence analysis of the entire capsid protein for a clear classification of the raccoon dog AstVs detected in Korea.

From studies of human AstVs, it is known that the AstV capsid protein is divided into a region encoding the core protein and a region encoding the spike protein [27]. In addition, the 5′ region encoding the core protein has been observed to be conserved among AstV strains clustered within the same *MAstV* group [28,29]. Analysis of the capsid protein sequences of raccoon dog AstVs revealed that the raccoon dog AstV strains detected in Korea, except for raccoon dog AstV 18-026I, shared significant amino acid sequence similarities in the core protein region with canine AstV strains reported in Hungary. These similarities were higher than those observed between canine AstV strains in the core protein region [28]. These results suggest a possible relationship between the raccoon dog AstV strains identified in Korea and canine AstV. In addition, ORF1a and ORF1b of raccoon dog AstV strains detected in China also show high nucleotide sequence similarity to canine AstV in China, suggesting the possibility of interspecies transmission between raccoon dogs and dogs [18,30]. However, there are no complete capsid protein sequence data for canine AstV in Korea, preventing further sequence comparisons. In addition, there is no definitive evidence of interspecies transmission between raccoon dogs and dogs to date. Therefore, studies on the interspecies transmission and pathogenicity of AstV between raccoon dogs and dogs appear necessary.

NT-AstV is known to infect animal brains and cause lesions such as non-suppurative encephalitis [12]. In addition, viral RNA can be detected in neurons by in situ hybridization (ISH) in brain tissue infected with NT-AstV [11]. However, the raccoon dog brain tissue in which AstV was detected in Korea was damaged during the transportation and storage of the sample, and we were unable to identify histopathological lesions in the brain tissue or detect viral RNA by ISH. Thus, we analyzed the genetic characteristics of the AstV strain identified in raccoon dog brain tissue to determine its potential as NT-AstV. 

Phylogenetic analysis of AstVs revealed that strains previously identified as NT-AstVs in animals to date all belong to the HMO clade [11]. In addition, analysis of the capsid protein amino acid sequences of NT-AstVs belonging to the HMO clade revealed a conserved Q(I/L)QxR(F/Y) motif sequence [12]. The raccoon dog AstV 17-148B strain detected in the brain tissue of raccoon dog was identified as belonging to the HMO clade, along with NT-AstVs previously detected in animals. Raccoon dog AstV 17-148B was also found to have a conserved Q(I/L)QxR(F/Y) motif sequence, suggesting a high potential for NT-AstV. 

But strains 17-153B, 17-157B, and 18-038B detected in the brain tissue of raccoon dogs did not belong to the HMO clade, and no conserved sequences such as the Q(I/L)QxR(F/Y) motif, were identified. However, previous studies have shown that porcine AstV (PoAstV) 2 and 5, but not PoAstV 3, known as NT-AstV, were detected in the brain tissue of piglets with congenital tremors [31]. There have also been reports of AstV, which appears to be of canine origin, being detected in the brain of crab-eating foxes with symptoms of central nervous system (CNS) disease [32]. The cases of 17-153B, 17-157B, and 18-038B in this study appear to be similar to PoAstV 2, 5 and crab-eating fox AstV, but the route of infection and pathogenicity of these AstV strains remains unclear. It is also possible that these AstV strains are NT-AstVs other than the HMO clade, such as NT-AstVs of the MLB clade identified in humans [33]. Therefore, further studies are needed to investigate the neurological manifestations and/or associated pathogenicity of these AstV strains.

In general, MAstV is found in mammals, while AAstV infects in birds. However, several studies have reported cases of AAstV, but not MAstV, being detected in the intestines or feces of mammalian species such as cats and mink [5,34]. Indeed, this was observed in mink individuals consuming chicken organs as their food [5]. Similarly, in humans, AstVs from other species of animals were detected in fecal samples, suggesting the possibility of AAstV transmission to mammals through contaminated food sources [35]. The 18-026I, detected in the intestinal tissue of raccoon dogs, showed the highest similarity to chicken AstV, which belongs to the group AAstV. These cases, similar to those observed in mink, suggest that raccoon dogs ingested contaminated food or other sources associated with chicken AstV, resulting in the detection of chicken-like AstV in their intestines. Further studies are needed to investigate the pathogenicity and interspecies transmission potential of exposure to these other species of AstV in raccoon dogs.

## 5. Conclusions

In conclusion, AstVs were detected in wild raccoon dogs living in Korea, with AstV detected in raccoon dog brain tissue for the first time. The analysis of AstV capsid protein sequences confirmed the potential for AstV cross-species transmission between raccoon dogs and dogs. Genetic analysis indicated the neurotropic AstV potential of these AstV strains detected in raccoon dog brain tissues. The continuous monitoring of AstV in raccoon dogs is essential to further investigate the transmission and pathogenicity of these viruses.

## Figures and Tables

**Figure 1 viruses-15-01488-f001:**
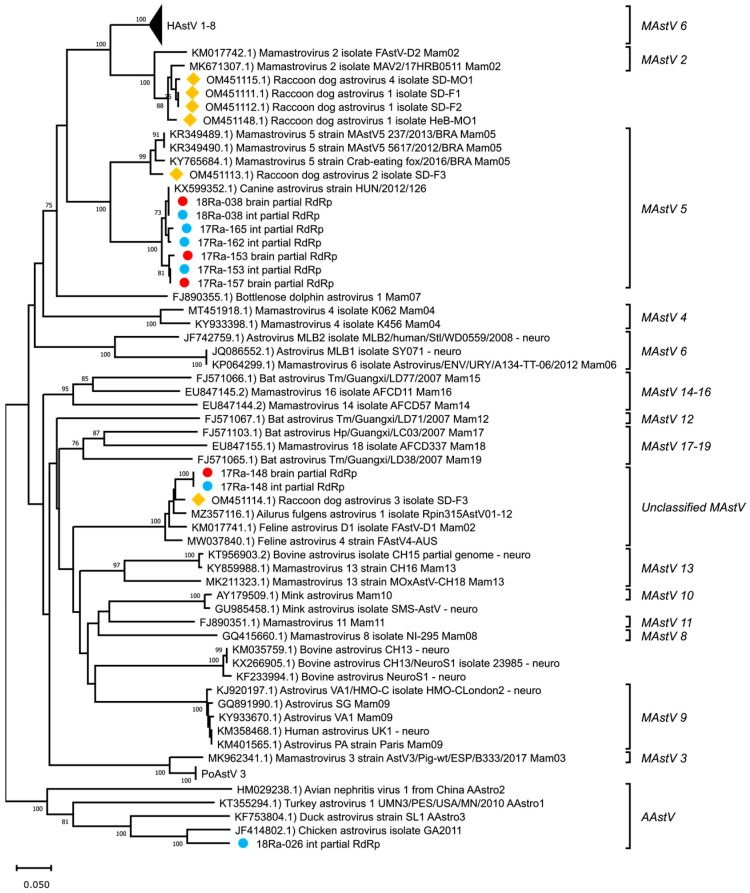
Phylogenetic analysis of partial RdRp nucleotide sequences of AstVs. The phylogenetic tree was generated using the neighbor-joining method with p-distance and bootstrapped with 1000 replications. Sequences of raccoon dog AstVs are labeled as follows: (●) detected in brain tissue, (●) detected in intestine tissue, (◆) reported in China.

**Figure 2 viruses-15-01488-f002:**
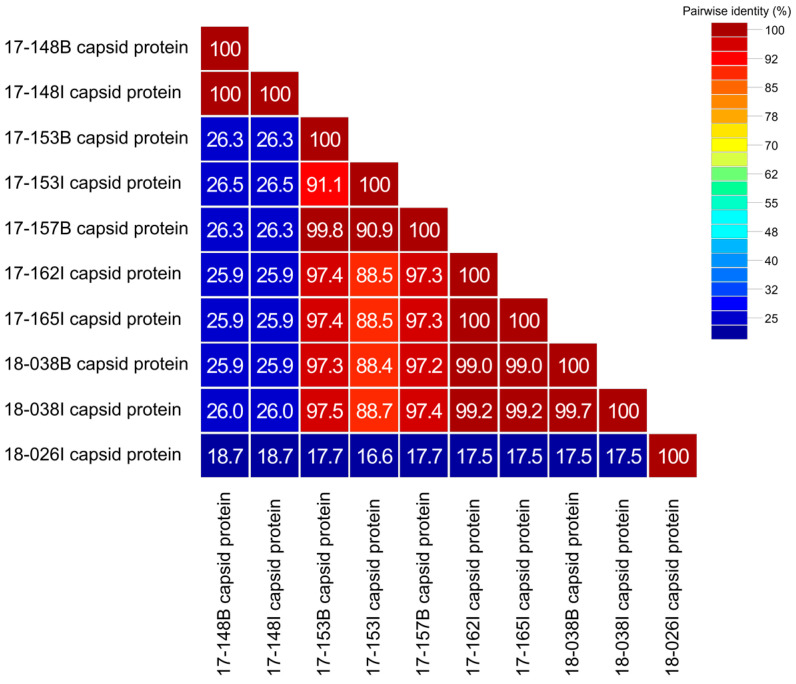
Capsid protein amino acid sequences identities between strains of raccoon dog AstVs detected in Korea.

**Figure 3 viruses-15-01488-f003:**
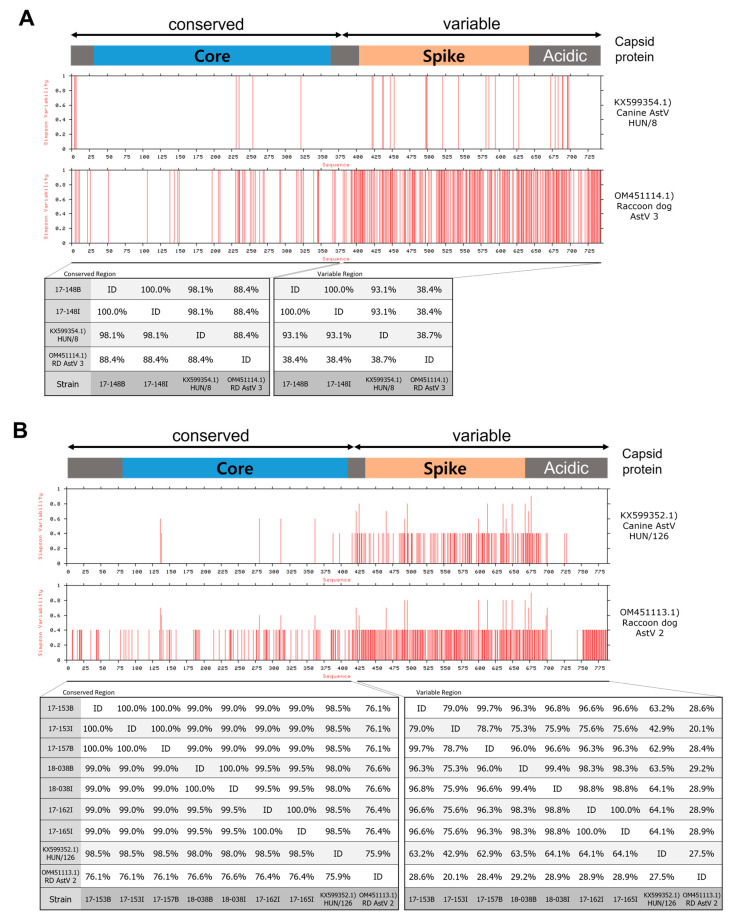
Variability and similarity of AstV capsid proteins between canine and raccoon dog strains. (**A**) Comparison of capsid proteins of 17-148B, 17-148I, canine AstV HUN/8, and raccoon dog AstV 3. (**B**) Comparison of capsid proteins between raccoon dog AstV strains belonging to MAstV5 and canine AstV HUN/126.

**Figure 4 viruses-15-01488-f004:**
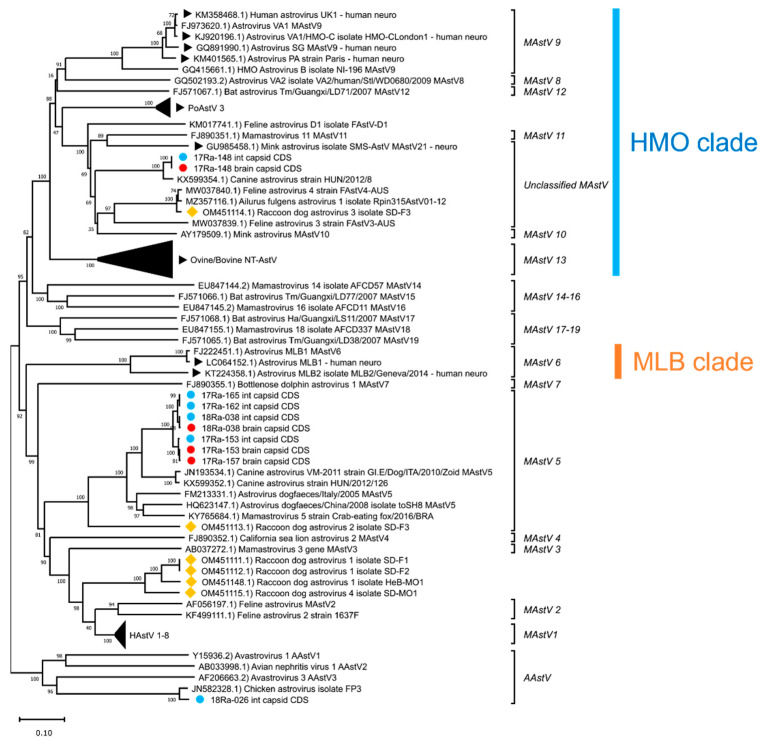
Phylogenetic analysis of complete capsid protein amino acid sequences of AstVs. The phylogenetic tree was generated using the neighbor-joining method with p-distance and bootstrapped with 1000 replications. Sequences of raccoon dog AstVs are labeled as follows: (**●**) detected in brain tissue, (**●**) detected in intestine tissue, (◆) reported in China. NT-AstVs are labeled with (▶).

**Figure 5 viruses-15-01488-f005:**
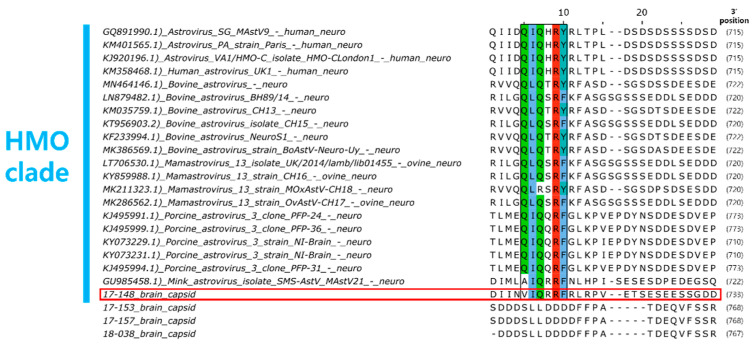
Conserved region of the Q(I/L)QxR(F/Y) motif in HMO clade NT-AstVs capsid protein amino acid sequences (marked in color). The red box indicates the raccoon dog AstV 17-148B strain detected in raccoon dog brain tissue in South Korea, which belongs to the HMO clade.

**Table 1 viruses-15-01488-t001:** PCR primers for raccoon dog AstV ORF2.

Primer Name	Position		Sequence	Reference
RAD AstV 148	RdRp	F1	TGG ATG AGC AAT ATC AGA CAC C	customized primer
RAD AstV 153			CAT CAA GAG GCT ACG CTG G	
RAD AstV 157				
RAD AstV 038			TAT CAA GAA ACT ACG CTG G	
RAD AstV 026			TAG CCT CAA AGT ATA AGA CGC A	
s2m_rev	ORF2	R	CCC TCG ATC CTA CTC GG	[23]
RAD AstV 148	RdRp	F2	CGA TTG GTA TTG TAA GAA CAT C	customized primer
RAD AstV 153			TGG TTA ATG CCG AGC AGC GGA A	
RAD AstV 157				
RAD AstV 038				
RAD AstV 026			ACA AGG GGT TGT TCG ATT G	

**Table 2 viruses-15-01488-t002:** Information of AstV-positive samples and prevalence in raccoon dogs.

Raccoon Dog No.	Case no.	Region	AstV Positive		Prevalence	
			Brain Tissue	Intestine Tissue	Brain	Intestine
17-148	KNU-082	Gyeonggi-do	+	+	3% (4/133)	7.8% (6/77)
17-153	KNU-087	Gyeonggi-do	+	+		
17-157	KNU-091	Gangwon-do	+	-		
17-162	KNU-096	Gangwon-do	-	+		
17-165	KNU-099	Gyeonggi-do	-	+		
18-026	CB-010	Chungcheong-do	-	+		
18-038	KNU-027	Gyeonggi-do	+	+		

**Table 3 viruses-15-01488-t003:** Partial RdRp nucleotide sequences similarity of raccoon dog AstVs detected in Korea.

Strain	Canine AstV Strain HUN 126 (KX599352.1)	*Ailurus fulgens* AstV 1 (MZ357116.1)	Chicken AstV Isolate GA2011 (JF414802.1)	Raccoon Dog AstV (China Strain)
17-148B	55.6%	90.1%	45.2%	54.6–89.3%
17-148I	55.6%	90.1%	45.2%	54.6–89.3%
17-153B	93.2%	55.1%	45.0%	55.6–74.4%
17-153I	94.2%	55.6%	46.0%	56.2–75.5%
17-157B	94.2%	55.6%	46.0%	56.2–75.5%
17-162I	95.1%	54.9%	45.2%	54.9–75.2%
17-165I	94.8%	54.6%	45.0%	54.6–75.0%
18-038B	97.5%	56.7%	45.7%	56.2–75.0%
18-038I	97.5%	56.7%	45.7%	56.2–75.0%
18-026I	45.2%	46.2%	76.7%	43.3–47.2%

**Table 4 viruses-15-01488-t004:** Capsid protein amino acid sequence similarity in raccoon dog AstVs detected in Korea.

Strain	Canine AstV Strain HUN/126 (KX599352.1)	Canine AstV Strain HUN/8 (KX599354.1)	Chicken AstV (JN582328.1)	Raccoon Dog AstV (China Strain)
17-148B	18.4%	91.4%	13.4%	17.2–59.1%
17-148I	18.4%	91.4%	13.4%	17.2–59.1%
17-153B	83.5%	19.3%	14.0%	18.3–49.8%
17-153I	74.9%	19.5%	13.2%	18.0–47.2%
17-157B	83.4%	19.3%	14.0%	18.3–49.8%
17-162I	83.5%	19.4%	14.1%	18.4–50.0%
17-165I	83.5%	19.4%	14.1%	18.4–50.0%
18-038B	83.0%	19.4%	14.0%	18.4–50.2%
18-038I	83.3%	19.5%	14.0%	18.4–50.1%
18-026I	13.9%	12.6%	96.3%	13.1–16.0%

## Data Availability

Not applicable.

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
