# Peer review of "Detection and Genetic Characterization of Astroviruses in Brain Tissues of Wild Raccoon Dogs"

_viruses, 2023, doi:10.3390/v15071488_

Round 1

Reviewer 1 Report

The paper I reviewed “Detection and genetic characterization of astroviruses in brain tissues of wild raccoon dogs” represents a study aimed to investigate the presenceand the genetic heterogeneity of astroviruses (AstVs) in intestine and brain of 133 wild racoon dogs in South Korea. AstVs RNA was detected in 7 animals of which 3 resulted positive either in intestine and brain, 3 only in intestine and 1 only in brain. Sequences analyses of 422 bp partial RdRp and complete capsid protein genes revealed for nine strains the highest identity for canine AstVs detected in Hungary, belonging to MAstV 5 (n. 7 sequences) and to an unclassified MAstV group (n.2 sequences), and for one sequence (18-026I) the highest identity was found to a chiken AstV (AAstV genus). Phylogenetic analysis revealed that of the four strains detected in brain tissue of raccoon dog, one was identified as belonging to the HMO clade in which grouped all the AstVs previously identified as neurotropic-AstVs, also demonstrated by the analysis of the capsid protein amino acid sequences of NT-AstVs belonging to the HMO clade that revealed a conserved motif sequence.

Comments

The article is well written and interesting. The statements described are supported by detailed presented data. However, the principal weakness of this work is represented by the lack of GenBank submission numbers of all the sequences identified that are necessary to give the correct scientific value to the study. Furthermore, I have some observations:

1)Line 26: I suggest to replace “belong” with “belonging”.

2)Line 31-34: I suggest to replace this sentence “Since its discovery in a child in 1975, AstVs have been detected in various hosts, including pigs, turkeys, cattle, chickens, mink, cats, and dogs and novel AstV continues to be detected in various species [3]. Recently, novel AstVs have been detected in some other animals, including reptiles, fish, and amphibians that are not birds or mammals [4].” as follows “Since its discovery in a child in 1975, AstVs have been detected in various hosts as pigs, turkeys, cattle, chickens, mink, cats, and dogs and novel AstVs continue to be detected in various animals [3], including reptiles, fish, and amphibians [4].”

3)Line 38-40: The Authors should better explain the taxonomical classification of AstVs.

4)Line 53-55: The sentence should be rewritten more clearly.

5)Line 60: “showed” is not appropriate.

6)Line 76-83: in this section, I suggest to add information on the clinical status of animal sampled or if not available I suggest to add this information anyway.

7)Line 120: I suggest to add the overall prevalence detected.

8)Line 129-133: Which is the identity of the sequences detected to each other?I suggest to add this information.

9)Line155-157: The sentence “The amino acid sequence of the capsid protein between AstVs detected in the brain and intestine of the same raccoon dog individual showed 100% identity between 17-148B and 17-148I, 91.1% between 17-153B and 17-153I, and 99.7% between 18-038B and 18-038I.” should be replaced as follows “The amino acid capsid protein sequence identities between AstVs detected in the brain and intestine of the same raccoon dog were 100% between 17-148B and 17-148I, 91.1% between 17-153B and 17-153I, and 99.7% between 18-038B and 18-038I.”

10)Line 290: “is infected” should be replaced with “infects”.

11)Line 297: The Authors should be more detailed in the sentence “…to chicken AstV, a type of AAstV. “.

Author Response

Comments and Suggestions for Authors

The paper I reviewed “Detection and genetic characterization of astroviruses in brain tissues of wild raccoon dogs” represents a study aimed to investigate the presenceand the genetic heterogeneity of astroviruses (AstVs) in intestine and brain of 133 wild racoon dogs in South Korea. AstVs RNA was detected in 7 animals of which 3 resulted positive either in intestine and brain, 3 only in intestine and 1 only in brain. Sequences analyses of 422 bp partial RdRp and complete capsid protein genes revealed for nine strains the highest identity for canine AstVs detected in Hungary, belonging to MAstV 5 (n. 7 sequences) and to an unclassified MAstV group (n.2 sequences), and for one sequence (18-026I) the highest identity was found to a chiken AstV (AAstV genus). Phylogenetic analysis revealed that of the four strains detected in brain tissue of raccoon dog, one was identified as belonging to the HMO clade in which grouped all the AstVs previously identified as neurotropic-AstVs, also demonstrated by the analysis of the capsid protein amino acid sequences of NT-AstVs belonging to the HMO clade that revealed a conserved motif sequence.

Comments

The article is well written and interesting. The statements described are supported by detailed presented data. However, the principal weakness of this work is represented by the lack of GenBank submission numbers of all the sequences identified that are necessary to give the correct scientific value to the study. Furthermore, I have some observations:

1)Line 26: I suggest to replace “belong” with “belonging”.

RESPONSE: We corrected the words in line 26 (Introduction Section).

2)Line 31-34: I suggest to replace this sentence “Since its discovery in a child in 1975, AstVs have been detected in various hosts, including pigs, turkeys, cattle, chickens, mink, cats, and dogs and novel AstV continues to be detected in various species [3]. Recently, novel AstVs have been detected in some other animals, including reptiles, fish, and amphibians that are not birds or mammals [4].” as follows “Since its discovery in a child in 1975, AstVs have been detected in various hosts as pigs, turkeys, cattle, chickens, mink, cats, and dogs and novel AstVs continue to be detected in various animals [3], including reptiles, fish, and amphibians [4].”

RESPONSE: The authors have revised the manuscript accordingly in line 31-33 (Introduction Section,).

3)Line 38-40: The Authors should better explain the taxonomical classification of AstVs.

RESPONSE: Thank you for reading carefully and sorry for the confusion. We corrected the sentence in line 37-38 (Introduction Section).

4)Line 53-55: The sentence should be rewritten more clearly.

RESPONSE: We understand the reviewer’s concern. However, the exact intestinal-to-brain route of infection of NT-AstVs is not known, so we have corrected the information that is currently estimated from animal cases. line 53-58 (Introduction Section).

5)Line 60: “showed” is not appropriate.

RESPONSE: Thank you for your careful reading. We have revised the manuscript accordingly in line 62 (Introduction Section)

6)Line 76-83: in this section, I suggest to add information on the clinical status of animal sampled or if not available I suggest to add this information anyway.

RESPONSE: We understand the reviewer’s concern. However, due to the nature of wildlife, they are often found dead, making it difficult to determine their specific clinical status, which is not described in this paper.

7)Line 120: I suggest to add the overall prevalence detected.

RESPONSE: The manuscript has been revised as requested in line 124-127 (Results Section).

8)Line 129-133: Which is the identity of the sequences detected to each other?I suggest to add this information.

RESPONSE: The authors revised more clearly in line 135-136 (Result Section 3.2). Furthermore, additional information was added in line 101 with reference(s) (Materials and Methods Section 2.3).

9)Line155-157: The sentence “The amino acid sequence of the capsid protein between AstVs detected in the brain and intestine of the same raccoon dog individual showed 100% identity between 17-148B and 17-148I, 91.1% between 17-153B and 17-153I, and 99.7% between 18-038B and 18-038I.” should be replaced as follows “The amino acid capsid protein sequence identities between AstVs detected in the brain and intestine of the same raccoon dog were 100% between 17-148B and 17-148I, 91.1% between 17-153B and 17-153I, and 99.7% between 18-038B and 18-038I.”

RESPONSE: The authors truly appreciate the kind review and suggestion. The manuscript has been revised as requested. line 161-163 (Result Section 3.3).

10)Line 290: “is infected” should be replaced with “infects”.

RESPONSE: We corrected the words in line 296

11)Line 297: The Authors should be more detailed in the sentence “…to chicken AstV, a type of AAstV. “.

RESPONSE: Thank you for reading carefully and sorry for the confusion. The authors revised more details in line 302-303 (Discussion Section)

Reviewer 2 Report

Viruses - Detection and genetic characterization of astroviruses in brain tissues of wild raccoon dogs

Summary: The authors used molecular techniques to detect and classify astroviruses from the brain and intestines of raccoon dogs. The authors showed strong sequence homology between several of these strains and canine astroviruses. Additionally, high sequence homology for one strain and chicken astrovirus was appropriately discussed. The manuscript shows likely species cross-over of astroviruses between dogs and raccoon dogs. Furthermore, the authors correctly interpreted the detection of astroviruses both in the brain and intestine and investigated this with molecular techniques when the brain was too autolyzed for direct detection methods. Finally, the manuscript is well-written with only minor grammatical errors.

Major

No major revisions.

Minor

Line 10 – AstVs is plural but should be made singular.

Line 14 – Positive in which sample?

Line 26 – Switch to plural over singular here

Line 46,47 – There has been recent work showing that porcine astrovirus 4 infects the respiratory epithelium of pigs.

Line 280 – The cited paper, reference 31, is a poor example of diagnostic investigation. In that paper, the authors detected an agent and associated it with a background histologic finding in the brain. Atypical porcine pestivirus (APPV) would be the leading differential for the described congenital tremors. APPV has since been experimentally reproduced to cause congenital tremors without a microscopic lesion. Detection does not equate causation as the authors of this manuscript correctly discussed with their own investigation.

I would avoid using this reference as supportive evidence that astroviruses that aren’t in the HMO clade and don’t have conserved motifs can infect the CNS.  

Well written article. 

Author Response

Comments and Suggestions for Authors

Viruses - Detection and genetic characterization of astroviruses in brain tissues of wild raccoon dogs

Summary: The authors used molecular techniques to detect and classify astroviruses from the brain and intestines of raccoon dogs. The authors showed strong sequence homology between several of these strains and canine astroviruses. Additionally, high sequence homology for one strain and chicken astrovirus was appropriately discussed. The manuscript shows likely species cross-over of astroviruses between dogs and raccoon dogs. Furthermore, the authors correctly interpreted the detection of astroviruses both in the brain and intestine and investigated this with molecular techniques when the brain was too autolyzed for direct detection methods. Finally, the manuscript is well-written with only minor grammatical errors.

Line 10 – AstVs is plural but should be made singular.

RESPONSE: We corrected the words in line 10 (Abstract Section)

Line 14 – Positive in which sample?

RESPONSE: Thank you for reading carefully and sorry for the confusion. We corrected the sentence more clearly in line 14-15 (Abstract Section)

Line 26 – Switch to plural over singular here

RESPONSE: We corrected the words in line 26 (Introduction Section)

Line 46,47 – There has been recent work showing that porcine astrovirus 4 infects the respiratory epithelium of pigs.

RESPONSE: The authors truly appreciate the kind review and suggestion. Thank you for your careful reading. However, We are aware of the points mentioned by the reviewer (Rahe et al., 2023), but we do not consider it appropriate to include them in this paper.

Line 280 – The cited paper, reference 31, is a poor example of diagnostic investigation. In that paper, the authors detected an agent and associated it with a background histologic finding in the brain. Atypical porcine pestivirus (APPV) would be the leading differential for the described congenital tremors. APPV has since been experimentally reproduced to cause congenital tremors without a microscopic lesion. Detection does not equate causation as the authors of this manuscript correctly discussed with their own investigation.

I would avoid using this reference as supportive evidence that astroviruses that aren’t in the HMO clade and don’t have conserved motifs can infect the CNS.

RESPONSE: Thank you for the detailed review, but the reference [31] is about the "detection of porcine astroviruses 2 and 4 in piglet brains", and we do not see anything related to APPV and what the reviewer claims. We considered the above reference to being relevant because it describes the detection of enteric AstV, not the previously known NT-AstV, in the brain.

Rahe, M.C., Michael, A., Piñeyro, P.E., Groeltz-Thrush, J., Derscheid, R.J., Zhai, S.-L., 2023. Porcine Astrovirus 4 Detection in Lesions of Epitheliotropic Viral Infection in the Porcine Respiratory Tract. Transboundary and Emerging Diseases 2023, 1-4.

Round 2

Reviewer 1 Report

In my opinioni the manuscript could be published in present form.